# ADAPTIVE TEST-TIME COMPUTE ALLOCATION VIA TRAINING-FREE DIFFICULTY PROXIES

## ABSTRACT

Large language models (LLMs) excel at complex tasks but incur prohibitive computational costs, particularly when using techniques like self-consistency that require multiple generation attempts. This paper addresses the challenge of adaptive test-time compute allocation. We propose a framework that leverages **training-free difficulty proxies** derived directly from the LLM generation process to distribute a fixed compute budget across the test queries, without requiring specialized training for the allocation mechanism. Our objective is to maximize the number of solved instances by dynamically allocating more compute to difficult instances and less to simpler ones, while adhering to a total budget constraint. We first introduce several training-free proxies and empirically demonstrate their effectiveness in estimating instance difficulty. We then design an adaptive allocation strategy guided by these proxies, which is theoretically grounded in a novel bandit formulation. Experiments across math (MATH, GSM8K), coding (LiveCodeBench), and Q&A (e.g., GPQA-Diamond) benchmarks demonstrate that our method significantly outperforms both uniform budget allocation and training-based allocation baselines, solving substantially more problems under identical budget constraints. This work presents a practical and readily deployable approach to enhance the resource efficiency of LLM inference for demanding reasoning tasks.

## 1 INTRODUCTION

Large Language Models (LLMs) have demonstrated remarkable capabilities in tackling complex reasoning tasks, pushing the frontiers of intelligence in domains like mathematical reasoning and code generation (Guo et al., 2025; Team et al., 2023). However, the pursuit of higher performance often necessitates intensive scaling of test-time compute (Brown et al., 2024; Snell et al., 2024; Wu et al., 2024), which allows models to *think more* during inference. Techniques such as self-consistency (Wang et al., 2023), which select the consensus generation given multiple candidate solutions, or Best-of-N (Cobbe et al., 2021), which returns the best generation guided by an external verifier, can dramatically improve inference performance. This is particularly crucial in automatically verifiable domains, such as math and code generation, where producing numerous diverse generations significantly increases the probability of finding a correct solution (Brown et al., 2024). Yet, these methods (Brown et al., 2024; Wang et al., 2023) typically apply a uniform allocation strategy, leading to unnecessary compute costs and thus being suboptimal.

Problem instances naturally vary in **difficulty** (Damani et al., 2025; Ren et al., 2021): some are solvable with a single attempt, while others demand extensive compute. Uniform allocation means wasting compute on easy instances and potentially under-allocating to difficult ones that could be solved with more compute. This inefficiency limits the practical deployment of LLM inference systems, especially when operating under a compute budget constraint, which motivates a need for *adaptive* test-time compute allocation.

To achieve effective adaptive allocation, a system must first estimate the difficulty of problem instances. This is where difficulty proxies become essential, quantifiable metrics that can predict how difficult an instance is likely to be for the LLM. While recent approaches have explored training specialized models (Manvi et al., 2024; Muennighoff et al., 2025) or probes to predict problem difficulty (Damani et al., 2025), these methods suffer from significant practical limitations. They require substantial labeled data, introduce additional model training overhead, and impose expensive

inference costs to deploy, defeating the goal of efficient compute allocation. This motivates a crucial question insight: what if we could leverage signals already present within the LLM's generation process, without requiring any auxiliary model training or data? **Training-free proxies** for difficulty estimation offer a fundamentally more efficient and scalable alternative. By extracting proxies directly from the LLM during inference (e.g., entropy, variance of gradient norms, or generation length), we can achieve on-the-fly difficulty estimation with minimal overhead. While some recent works have begun exploring individual proxies such as generation length (Xue et al., 2025) or entropy (Yong et al., 2025), a systematic investigation of their effectiveness and relative performance for test-time compute allocation in LLMs remains largely unexplored. This leads us to our central research question:

> *What signals of problem difficulty do LLMs provide, and how to efficiently allocate a test-time compute budget based on these signals?*

To address this central question, this paper introduces a principled framework and a novel solution for adaptive test-time compute allocation. Our method begins with a rigorous empirical investigation into a diverse set of training-free difficulty proxies that are meticulously selected and adapted from prior work or newly proposed. We systematically evaluate their efficacy by measuring their correlation with an oracle measure of difficulty, revealing strong predictive capabilities for several candidates. Building upon these empirically validated proxies, we then formulate the adaptive compute allocation task as a specialized multi-armed bandit (MAB) problem with arm elimination upon success. Under this novel MAB framework, we develop our core contribution: DIPA (Difficulty-Informed Probabilistic Allocation), a theoretically grounded policy that intelligently navigates the exploration-exploitation trade-off by probabilistically allocating computational resources. Comprehensive experiments on challenging math, code, and Q&A benchmarks subsequently demonstrate that DIPA significantly outperforms standard uniform budget allocation and other heuristic strategies, thereby validating the efficacy of our integrated framework.

This work makes the following key contributions:

- We systematically define and validate diverse training-free difficulty proxies for adaptive test-time compute allocation, establishing their efficacy in estimating instance difficulty.
- We propose a novel reformulation of Test-Time Compute Allocation as a specialized Multi-Armed Bandit (MAB) problem with arm elimination upon success, providing the **first** MAB-based framework for this LLM inference challenge.
- We introduce DIPA, a novel allocation algorithm that strategically balances exploration and exploitation through probabilistic allocation based on dynamically updated difficulty estimates.
- We provide theoretical analysis showing DIPA's regret bound is highly related to difficulty proxy quality, highlighting the importance of selecting effective proxies.
- Through extensive experiments, we demonstrate DIPA consistently outperforms established baselines on both verifiable and non-verifiable domains, confirming its practical efficacy.

## 2 RELATED WORKS

We discuss the most related works here and extend our discussions in Appx. B.

**Adaptive Test-Time Compute Allocation for LLMs.** Several approaches have explored adaptive compute allocation. For LLM voting systems, existing methods include step-by-step sampling with early stopping criteria based on consistency scores (Aggarwal et al., 2023), as well as LLM-based filtering of generations introduced by (Chen et al., 2024a; Wang et al., 2025). However, these methods often depend on posterior estimations that struggle under tight budgets, simplify difficulty to a binary classification (easy or hard), or incur significant overhead from powerful ranking models (e.g., GPT-4). DIPA instead starts with a prior difficulty estimation, establishes fine-grained difficulty ranking, and does not require auxiliary models. Other recent, training-based methods include (Damani et al., 2025), who trained an LLM-based probe to predict the marginal benefit of additional compute, and (Zhang et al., 2025b), who estimated the success probability by learning from training data. In contrast, our DIPA framework provides a **training-free** adaptive allocation strategy, guided by the dynamically updated difficulty proxies.

**Instance Difficulty Estimation.** Prior training-based methods (Ren et al., 2021; Cui et al., 2023; Liu et al., 2024; Xue et al., 2025; Damani et al., 2025) often rely on training auxiliary difficulty

estimators, which require extensive labeled data and training compute. In classical machine learning, several training-free metrics have been explored, including entropy (Simsek et al., 2022; Huang et al., 2024; Yong et al., 2025), Variance of Gradients (VoG) (Agarwal et al., 2022), gradient norms inspired by Out-of-Distribution detection (Huang et al., 2021), and ensemble consistency (Jiang et al., 2021; Baldock et al., 2021). However, the application of these *training-free* proxies to test-time compute allocation in LLMs, and their effective adaptation, remained largely unstudied in LLMs. Our research presents the first systematic evaluation and utilization of such *training-free* proxies, derived directly from the LLM input or generation, specifically for adaptive test-time compute allocation, without auxiliary model training or extensive labeled data for the difficulty estimation mechanism.

## 3 PROBLEM FORMULATION

We address adaptive test-time compute allocation for LLMs. Given a set of $N$ instances $\mathcal{X} = \{\mathbf{x}_i\}_{i=1}^{N}$, we consider a total computational budget of $T$. Let $T_i \in \mathbb{N}_0$ (where $\mathbb{N}_0 = \mathbb{N} \cup \{0\}$ includes the possibility of allocating zero budget) be the budget allocated to process instance $\mathbf{x}_i$. The vector of allocations is denoted by $\boldsymbol{T} = (T_1, \ldots, T_N)$. Specifically, the budget $T_i$ for $\mathbf{x}_i$ is interpreted as the number of generation attempts made for this instance. We consider an indicator function $\mathbf{1}(\mathbf{o} \mid \mathbf{x}_i)$ which decides if any single generation $\mathbf{o}$ solves the instance $\mathbf{x}_i$. In a verifiable task, $\mathbf{1}(\mathbf{o} \mid \mathbf{x}_i)$ is its automatic oracle verifier. We discuss the indicator function for non-verifiable tasks in Sec. 5.4.

Our target is to find an allocation strategy that maximizes the total number of instances solved. Let $F(\mathbf{x}_i; T_i)$ be a meta indicator function that is 1 if instance $\mathbf{x}_i$ is solved within its $T_i$ allocated attempts and 0 otherwise. The objective is to maximize the **coverage** (i.e., the fraction of solved instances):

$$\max \frac{1}{N} \sum_{i=1}^{N} F(\mathbf{x}_i; T_i) \quad \text{s.t.} \sum_{i=1}^{N} T_i = T \ . \tag{1}$$

The core challenge addressed in this work is to determine the sequence of per-instance generation attempts $\boldsymbol{T}$. This determination relies on readily available, training-free proxies of instance difficulty, without necessitating model fine-tuning or specialized training for the allocation mechanism itself.

## 4 TRAINING-FREE PROXIES FOR DIFFICULTY ESTIMATION

### 4.1 TRAINING-FREE PROXIES

To begin with, we first introduce the following training-free proxies that we aim to study throughout this paper. Formally, given an input instance $\mathbf{x} = (x_1, \cdots, x_S)$, we use LLM to produce $m$ generations $\mathcal{O} \triangleq \{\mathbf{o}^{(i)} = (o_1^{(i)}, \cdots, o_{L_i}^{(i)})\}_{i=1}^{m}$ conditioned on $\mathbf{x}$ for the computation of the training-free proxy. We denote $\mathcal{Y} \triangleq \{\mathbf{y}^{(i)}\}_{i=1}^{m}$ as the corresponding final answers extracted from $\mathcal{O}$, and define the cross-entropy loss on the concatenated sequence $\mathbf{x} \oplus \mathbf{o}$ with next token prediction as $\text{CE}(\mathbf{x} \oplus \mathbf{o})$ with formal definition in Appx. E.3. Intuitively, an instance is difficult if: (1) the question or the reasoning process is long (Muennighoff et al., 2025), (2) the LLM is uncertain (Huang et al., 2024), (3) its prediction is sensitive to input perturbations (Agarwal et al., 2022; Huang et al., 2021), and (4) there is no obvious consensus within candidate solutions (Baldock et al., 2021). Inspired by similar intuitions from previous works (in Sec. 2), we formally introduce several training-free difficulty proxies for LLMs in inference:

**Question Length:** $\mathcal{M}_{\text{QL}}(\mathbf{x}) = |\mathbf{x}| = S$, **Entropy:** $\mathcal{M}_{\text{Ent}}(\mathcal{O} \mid \mathbf{x}) = \frac{1}{m} \sum_{i=1}^{m} \text{CE}(\mathbf{x} \oplus \mathbf{o}^{(i)})$,

**Gradient Norm:** $\mathcal{M}_{\text{GN}}(\mathcal{O} \mid \mathbf{x}) = \frac{1}{m} \sum_{i=1}^{m} \text{Mean}\{\|\nabla_x \text{CE}(\mathbf{x} \oplus \mathbf{o}^{(i)})\| \mid x \in \mathbf{x} \oplus \mathbf{o}\}$,

**Variance of Gradient:** $\mathcal{M}_{\text{VoG}}(\mathcal{O} \mid \mathbf{x}) = \frac{1}{m} \sum_{i=1}^{m} \text{Var}\{\|\nabla_x \text{CE}(\mathbf{x} \oplus \mathbf{o}^{(i)})\| \mid x \in \mathbf{x} \oplus \mathbf{o}\}$,

**Generation Consistency:** $\mathcal{M}_{\text{GC}}(\mathcal{O} \mid \mathbf{x}) = \frac{1}{m} \max_{c \in \mathcal{Y}} \sum_{i=1}^{m} \mathbb{I}[\mathbf{y}^{(i)} = c]$, and

**Generation Length:** $\mathcal{M}_{\text{GL}}(\mathcal{O} \mid \mathbf{x}) = \frac{1}{m} \sum_{i=1}^{m} |\mathbf{o}^{(i)}| = \frac{1}{m} \sum_{i=1}^{m} L_i$.

We denote the input-based and generation-based proxy as $\mathcal{M}(\mathbf{x})$ and $\mathcal{M}(\mathcal{O} \mid \mathbf{x})$, respectively. For formulations of other input-based proxies $\mathcal{M}_{\text{Ent}}(\mathbf{x})$, $\mathcal{M}_{\text{GN}}(\mathbf{x})$, and $\mathcal{M}_{\text{VoG}}(\mathbf{x})$, refer to Appx. E.3.

Table 1: Spearman correlations between proxies and $\text{Pass@}k^{-1}(\tau)$. Values are reported in mean from 3 trials with $m = 3$ (all std $< 0.03$). The input-based proxies (without generation) are highlighted in  purple cell , while generation-based proxies are unshaded. The highest absolute correlation is highlighted in **bold**, and the second highest is underlined.

| Difficulty Proxy | MATH500 | | GSM8K | | LiveCodeBench |
|---|---|---|---|---|---|
| | QM-1.5B | Llama-8B | QM-1.5B | Llama-8B | Llama-8B |
| Level (1-5) | 0.488 | 0.515 | | | |
| $\mathcal{M}_{\text{Ent}}(\mathbf{x})$ | 0.496 | 0.457 | 0.088 | 0.160 | 0.244 |
| $\mathcal{M}_{\text{GN}}(\mathbf{x})$ | $-0.489$ | $-0.443$ | $-0.085$ | $-0.292$ | $-0.430$ |
| $\mathcal{M}_{\text{VoG}}(\mathbf{x})$ | $-0.468$ | $-0.464$ | $-0.045$ | $-0.316$ | $-0.488$ |
| $\mathcal{M}_{\text{QL}}(\mathbf{x})$ | 0.482 | 0.450 | 0.382 | 0.302 | 0.502 |
| $\mathcal{M}_{\text{Ent}}(\mathcal{O} \mid \mathbf{x})$ | 0.180 | 0.086 | 0.454 | 0.301 | 0.373 |
| $\mathcal{M}_{\text{GN}}(\mathcal{O} \mid \mathbf{x})$ | $-0.311$ | $-0.530$ | 0.276 | $\underline{-0.565}$ | $-0.487$ |
| $\mathcal{M}_{\text{VoG}}(\mathcal{O} \mid \mathbf{x})$ | $-0.286$ | $-0.496$ | 0.226 | $-0.555$ | $\mathbf{-0.567}$ |
| $\mathcal{M}_{\text{GC}}(\mathcal{O} \mid \mathbf{x})$ | $\underline{-0.677}$ | $\underline{0.651}$ | $\mathbf{-0.651}$ | $\mathbf{-0.663}$ | $-0.213$ |
| $\mathcal{M}_{\text{GL}}(\mathcal{O} \mid \mathbf{x})$ | $\mathbf{0.780}$ | $\mathbf{0.701}$ | $\underline{0.592}$ | 0.467 | $\underline{0.530}$ |

## 4.2 EMPIRICAL STUDY OF DIFFICULTY ESTIMATION

**Oracle Difficulty.** To empirically evaluate the effectiveness of the training-free difficulty proxies introduced above on the estimation of instance difficulty, we first establish a ground-truth measure of intrinsic instance difficulty, termed the oracle difficulty measure. This oracle quantifies the minimum number of generations required to achieve a predefined target probability of success $\tau \in [0, 1]$ in solving a given problem instance. We define this measure using the inverse of the standard $\text{Pass@}k$ metric, denoted $\text{Pass@}k^{-1}(\tau)$. The $\text{Pass@}k$ metric itself computes the probability of obtaining at least one correct solution when drawing $k$ samples without replacement from a finite pool of $K$ available generations, of which $K^+$ are correct. Mathematically, for $k \leq K$, $\text{Pass@}k$ (Chen et al., 2021) is given by:

$$\text{Pass@}k = 1 - \frac{\binom{K - K^+}{k}}{\binom{K}{k}} . \tag{2}$$

The oracle difficulty $\text{Pass@}k^{-1}(\tau)$ is then the smallest positive integer $k$ such that $\text{Pass@}k \geq \tau$:

$$\text{Pass@}k^{-1}(\tau) \triangleq \min\{k \in \mathbb{N}^+ \mid \text{Pass@}k \geq \tau\} . \tag{3}$$

Intuitively, a lower value of $\text{Pass@}k^{-1}(\tau)$ indicates an easier instance, as fewer generations are needed to reach the success threshold $\tau$, while a higher value signifies greater difficulty. This oracle, therefore, provides a principled benchmark against which the correlation and utility of training-free difficulty proxies can be rigorously assessed.

**Correlation Evaluation.** To empirically validate our training-free difficulty proxies, we examine their Spearman rank correlation with an oracle difficulty measure ($\text{Pass@}k^{-1}(\tau)$). For math benchmarks, MATH500 (Lightman et al., 2023) and GSM8K (Cobbe et al., 2021)), we analyze the correlation on a math-specific LLM Qwen2.5-Math-1.5B (Yang et al., 2024) and a general LLM Llama3.1-8B (Grattafiori et al., 2024). For code generation benchmark, LiveCodeBench (Jain et al., 2025), we analyze on Llama3.1-8B only. The results in Tab. 1 indicate that proxies derived from the generation process of LLMs, e.g., Generation Length $\mathcal{M}_{\text{GL}}$, VoG $\mathcal{M}_{\text{VoG}}$, and Generation Consistency $\mathcal{M}_{\text{GC}}$, exhibit robust correlations and excel in different tasks. Specifically, $\mathcal{M}_{\text{GL}}$ performs the best on MATH500 for both models, $\mathcal{M}_{\text{GC}}$ performs the best on GSM8K for both models, and $\mathcal{M}_{\text{VoG}}$ is the best proxy on LiveCodeBench. This affirms their utility in guiding instance-aware and task-specific compute allocation. Among all evaluated metrics, $\mathcal{M}_{\text{GL}}$ consistently emerges as a particularly compelling proxy due to its simplicity and strong empirical performance (correlations are always greater than 0.467). See more discussions in Appx. F.1.

Intriguingly, certain input-based proxies (detailed in Appx. E.3) also prove effective, offering a valuable, low-cost initial difficulty estimate *even before any generation occurs*, which can serve as a

prior for adaptive allocation (see Sec. 5.2). Regarding proxies informed by LLM generations $\mathcal{O}$, their estimation quality intuitively improves with the sample size $m$, yet our result in Fig. 5 in Appx. F.1 shows that marginal gains diminish rapidly. A relatively small $m$ (e.g., $m = 4$) often suffices for strong difficulty estimation, highlighting the cost-effectiveness of these dynamic proxies.

In summary, our extensive correlation analysis confirms that readily available, training-free signals, whether derived from the input instance itself or the generation process of LLMs, provide potent and efficient means to estimate instance difficulty. The strong performances of $\mathcal{M}_{\mathrm{GL}}$, $\mathcal{M}_{\mathrm{GC}}$, and $\mathcal{M}_{\mathrm{VoG}}$, showcase those efficient training-free proxies that can significantly inform adaptive compute allocation strategies for specialized tasks, paving the way for more resource-aware LLM inference.

## 5  ADAPTIVE TEST-TIME COMPUTE ALLOCATION VIA TRAINING-FREE PROXIES

Based on our comprehensive study of difficulty proxies in Sec. 4.2, we propose a novel approach for adaptive test-time compute allocation with LLMs in this section. We reformulate this problem as a specialized multi-arm bandit (MAB) variant featuring arm elimination upon success and establish the first general MAB-based framework to address it (Sec. 5.1). Within this framework, we introduce DIPA (Difficulty-Informed Probabilistic Allocation) as an effective and efficient solution (Sec. 5.2).

### 5.1  REFORMULATION AS MULTI-ARMED BANDIT VARIANT

We reformulate the adaptive test-time compute allocation problem as a stochastic multi-armed bandit (MAB) variant characterized by a global budget, arm elimination upon success, and instance-specific reward dynamics dependent on cumulative interaction. Formally, let the set of $N$ instances $\mathcal{X} = \{\mathbf{x}_i\}_{i=1}^{N}$ constitute the set of available arms. The process unfolds over a maximum of $T$ discrete rounds, where $T$ is the total computational budget. Each round corresponds to a single pull of an arm, consuming one unit of budget. The state of each arm $\mathbf{x}_i$ is defined by $(s_i, T_i)$, where $s_i \in \{0, 1\}$ is its current status (0 for unsolved, 1 for solved) and $T_i$ is the cumulative budget (number of pulls) allocated to arm $\mathbf{x}_i$. Initially, $s_i = 0$ and $T_i = 0$ for all $i \in [N]$. We then introduce the following general framework to solve this MAB reformulation:

---

**A General MAB-Based Framework for Adaptive Test-Time Compute Allocation**

In each round $t \in [T]$:
  (I)   A policy $\pi$ selects an arm $\mathbf{x}_j$ from the set of unsolved arms $\{\mathbf{x}_i \in \mathcal{X} \mid s_i = 0\}$.
  (II)  The budget allocated to $\mathbf{x}_j$ is incremented: $T_j \leftarrow T_j + 1$.
  (III) An outcome $F(\mathbf{x}_j; T_j)$ is observed. If $F(\mathbf{x}_j; T_j) = 1$, the status of arm $\mathbf{x}_j$ transitions to solved ($s_j \leftarrow 1$).
  (IV)  $r_k = 1$ if an arm transitions from status 0 to 1 in round $t$; otherwise, $r_k = 0$.

---

The objective is then to design a policy $\pi$ that maximizes the total number of unique arms solved within the $T$ available pulls:

$$\max_{\pi} \sum_{t=1}^{T} r_k = \max_{\pi} \sum_{i=1}^{N} s_i \,. \tag{4}$$

Since the performance function $F(\mathbf{x}_i; T_i)$ in the original problem is interpreted as yielding a deterministic binary indicator of success (1 if solved, 0 otherwise) for instance $\mathbf{x}_i$ given $T_i$ units of budget, then the optimization in Eq. 1 is equivalent to maximizing the sum in Eq. 4. In this bandit reframing, $T_i$ (i.e., the cumulative pulls for arm $\mathbf{x}_i$ after $T$ rounds under policy $\pi$) directly corresponds to the per-instance budget $T_i$ in the original problem. The policy $\pi$ makes sequential decisions over up to $T$ rounds, and the set of final cumulative pulls $\{T_i(\pi)\}_{i=1}^{N}$ forms an allocation $\boldsymbol{T}_\pi$ such that $\sum T_i(\pi) = T$. An optimal policy $\pi^*$ for Eq. 4 therefore identifies an allocation $\boldsymbol{T}_{\pi^*}$ that maximizes the number of successfully processed instances (those with $s_i = 1$), directly addressing the aim of the original problem under this interpretation of $F$ as a deterministic, binary success function.

---

**Algorithm 1** Difficulty-Informed Probabilistic Allocation (`DIPA`)

---

1: **Input:** Compute budget $T$, instance set $\mathcal{X} = \{\mathbf{x}_i\}_{i=1}^N$, input-based proxy $\mathcal{M}_{\text{input}}$, generation-based proxy $\mathcal{M}_{\text{gen}}$
2: **Initialization:** For each instance $\mathbf{x}_i \in \mathcal{X}$: $\mathcal{M}_i \leftarrow \mathcal{M}_{\text{input}}(\mathbf{x}_i)$ and $\mathcal{O}_i \leftarrow \varnothing$
3: **for** each compute allocation step $t \in [T]$ **do**
4:    **if** $\mathcal{X} = \varnothing$ **then**
5:       **break** // All instances solved, stop
6:    **end if**
7:    Update sampling probabilities: $P_k \leftarrow \frac{1/\mathcal{M}_k^\lambda}{\sum_{\mathbf{x}_l \in \mathcal{X}} 1/\mathcal{M}_l^\lambda}$ for all $\mathbf{x}_k \in \mathcal{X}$
8:    Sample $\mathbf{x}_j$ from $\mathcal{X}$ using probabilities $\{P_k\}$
9:    Produce generation $\mathbf{o}_j^{(t)}$ for $\mathbf{x}_j$, update $\mathcal{O}_j \leftarrow \mathcal{O}_j \cup \{\mathbf{o}_j^{(t)}\}$, and verify $\mathbf{1}(\mathbf{o}_j^{(t)} \mid \mathbf{x}_j)$
10:   **if** $\mathbf{1}(\mathbf{o}_j^{(t)} \mid \mathbf{x}_j) = 1$ **then**
11:      $\mathcal{X} \leftarrow \mathcal{X} \setminus \{\mathbf{x}_j\}$ // Remove solved instance
12:   **else**
13:      Update $\mathcal{M}_j \leftarrow \mathcal{M}_{\text{gen}}(\mathcal{O}_j \mid \mathbf{x}_j)$ // Update difficulty estimate
14:   **end if**
15: **end for**
16: **Output:** The final set of (unsolved) instances $\mathcal{X}$.

---

## 5.2 Difficulty-Informed Probabilistic Allocation

Building on this multi-armed bandit (MAB) reformulation, we now propose a policy $\pi$ that strategically leverages training-free proxies of instance difficulty to determine the arm selection process. This policy is detailed in Algo. 1. The fundamental principle is to probabilistically prioritize arms (instances) estimated to be easier, thereby aiming to maximize the count of successfully resolved instances (i.e., $s_i = 1$) within the allocated budget $T$, while concurrently permitting exploration of instances that might have been erroneously classified as more difficult. This methodology directly confronts the challenge of determining per-instance allocations $T_i$ without necessitating specialized training for the allocation mechanism itself.

Let $\mathcal{M}_i$ represent the difficulty estimate assigned to instance (arm) $\mathbf{x}_i$, initialized by an input-based proxy measure $\mathcal{M}_i \leftarrow \mathcal{M}_{\text{input}}(\mathbf{x}_i)$ (line 2 of Algo. 1). Conventionally, a higher value of $\mathcal{M}_i$ signifies a more challenging instance. Our target is to construct a selection probability for each currently unsolved arm $\mathbf{x}_i$ that exhibits an inverse relationship with its estimated difficulty $\mathcal{M}_i$. In each round $t \in \{1, \dots, T\}$, the set of active (unsolved) instances is denoted by $\mathcal{X}$, which directly corresponds to the MAB concept of $\mathcal{U}_t = \{\mathbf{x}_i \mid \mathbf{x}_i \text{ is unsolved at step } t\}$. The policy selects an arm $\mathbf{x}_j \in \mathcal{X}$ for the subsequent budget allocation (i.e., a generation attempt) based on a probability distribution defined over this set $\mathcal{X}$. This distribution is formulated as follows: For every unsolved arm $\mathbf{x}_k \in \mathcal{X}$, its selection probability $P_k$ is updated (line 7 of Algo. 1) based on its current difficulty proxy $\mathcal{M}_k$:

$$P_k = \text{Prob}(\text{select } \mathbf{x}_k \text{ in step } t \mid \mathcal{X}, \{\mathcal{M}_l\}_{\mathbf{x}_l \in \mathcal{X}}) = \frac{w_k}{\sum_{\mathbf{x}_l \in \mathcal{X}} w_l} \tag{5}$$

where $w_k = 1/\mathcal{M}_k^\lambda$ is the sample proxy weight and $\lambda \propto |\mathcal{X}|$ is an active sampling temperature.

Upon the selection of arm $\mathbf{x}_j$, a generation $\mathbf{o}_j^{(t)}$ is produced and is added to its set of generations $\mathcal{O}_j$ (line 9 of Algo. 1). This step consumes one unit of the compute budget for instance $\mathbf{x}_j$. Subsequently, the correctness of this generation is verified, yielding an outcome $\mathbf{1}(\mathbf{o}_j^{(t)} \mid \mathbf{x}_j)$. If this outcome indicates success (e.g., $\mathbf{1}(\mathbf{o}_j^{(t)} \mid \mathbf{x}_j) = 1$), instance $\mathbf{x}_j$ is considered solved and is removed from the set of active instances, i.e., $\mathcal{X} \leftarrow \mathcal{X} \setminus \{\mathbf{x}_j\}$ (line 11 of Algo. 1). Conversely, if the instance $\mathbf{x}_j$ is not solved ($\mathbf{1}(\mathbf{o}_j^{(t)} \mid \mathbf{x}_j) \neq 1$), it remains in the set $\mathcal{X}$ for subsequent steps. Importantly, as shown in Algo. 1 (line 13), if arm $\mathbf{x}_j$ is not solved, its difficulty proxy $\mathcal{M}_j$ is re-evaluated and updated using a generation-based proxy $\mathcal{M}_{\text{gen}}(\mathcal{O}_j \mid \mathbf{x}_j)$ (e.g., the generation length $\mathcal{M}_{\text{GL}}$ in Sec. 4.1) as generation-based proxies usually achieve higher correlations with the oracle difficulty (see Sec. 4.2). This dynamic update of $\mathcal{M}_j$ allows the policy to adapt its estimate of instance difficulty based on the interaction history (the set of generations $\mathcal{O}_j$), refining future selection probabilities. The probabilities $\{P_k\}$ will be re-calculated at the beginning of the next iteration based on the potentially updated $\mathcal{X}$ and $\mathcal{M}_i$ values. If all instances are solved ($\mathcal{X} = \varnothing$), the process terminates (lines 4-6 of Algo. 1).

This probabilistic allocation policy, informed by a dynamically updated difficulty estimate, presents several compelling advantages. *(a)* It facilitates an adaptive allocation of test-time compute, dynamically redirecting effort towards instances offering a higher success rate as other instances are progressively solved or as difficulty estimates are refined. *(b)* Crucially, the allocation policy itself is **training-free**. *(c)* The probabilistic nature of the selection mechanism inherently balances exploitation of perceived easy instances with exploration of those deemed harder, mitigating the risk of prematurely abandoning instances that might be solvable. Consequently, this approach can lead to enhanced efficiency, potentially achieving a higher coverage for a given total budget $T$.

### 5.3 THEORETICAL ANALYSIS OF REGRET BOUND FOR DIPA

To theoretically justify DIPA, we state a concise inversion-based regret bound for DIPA under a static proxy. Refer to Appx. D for full proof, detailed setups, and more discussions.

**Theorem 1** (Regret bound via Kendall–tau inversions). *Consider $N$ arms with unknown success probabilities $p_1, \ldots, p_N \in [0, 1]$, ordered so that $p_1 \geq \cdots \geq p_N$. Let a static proxy induce positive weights $w_1, \ldots, w_N$. Define the Kendall–tau inversion set $\mathrm{Inv} := \{(i, j) : i < j \text{ and } w_i < w_j\}$ and $K := |\mathrm{Inv}|$. Assume the **proxy quality** condition: $\exists \gamma \geq 1 : \forall i, j \in \mathbb{N}, \; \frac{1}{\gamma} \leq \frac{w_i/p_i}{w_j/p_j} \leq \gamma$. Then the cumulative regret $R(T)$, against the optimal policy (selects arm $\arg\max_{k \in \mathcal{X}} p_k$), satisfies*

$$R(T) \; \leq \; \sum_{(i,j) \in \mathrm{Inv}} \gamma \, \frac{(p_i - p_j)}{p_i} \; \leq \; \gamma K.$$

Thm. 1 shows that the cumulative regret is controlled by the proxy quality $\gamma$ and the size of the Kendall–tau inversion set $|\mathrm{Inv}|$. Specifically, Inv captures the pairwise ranking errors between the proxy-induced order and the oracle order of arms. Each element $(i, j) \in \mathrm{Inv}$ corresponds to a misordered pair where the proxy ranks the weaker arm $j$ above the stronger arm $i$.

**Remarks on Proxy Quality Assumption.** The performance of DIPA depends on the quality of the difficulty proxies. We formalize proxy quality with a multiplicative error assumption, which bounds how much the proxy-based weights can deviate from the true success probabilities. A larger $\gamma$ indicates a poorer proxy that distorts the relative weights of arms more significantly, leading to errors in the sampling probability $P_k$.

**Remarks on Regret Bound.** The bound ties the $R(T)$ to the ranking error of the proxy: small $K$ (high rank correlation) yields small regret, independent of the budget $T$; when $\gamma = 1$ (perfect proxy alignment to oracle $p$), the bound collapses to zero with $K = 0$. This aligns with our empirical findings in Sec. 6 where difficulty proxies with higher correlations used in DIPA generally perform better, which also validates our assumption on proxy quality.

### 5.4 APPLICATION OF DIPA IN NON-VERIFIABLE TASKS

In non-verifiable tasks, we propose integrating an external reward model $\mathrm{RM}(\cdot): \mathcal{O} \to \mathbb{R}$ to define the verification function $\mathbf{1}_{\mathrm{RM}}(\mathbf{o}^{(t)} \mid \mathbf{x}) := 1$ if $\mathrm{RM}(\mathbf{o}^{(t)}|\mathbf{x}) \geq \mathrm{RM}(\mathbf{o}^{(t-j)}|\mathbf{x}) \; \forall j \in [1, C]$ else 0. Namely, the reward model evaluates generations iteratively and eliminates $\mathbf{x}$ when no higher-reward generation is found for $C$ consecutive attempts, indicating the instance is likely solved. In inference, only the highest-reward generation for each instance is evaluated (if actually solved).

## 6 EXPERIMENTS

This section empirically validates our DIPA. We first present main results demonstrating the superiority of DIPA over baselines in maximizing coverage under fixed budgets (Sec. 6.1), then perform ablation studies to validate the design choices of DIPA, offering insights into its performance and the efficacy of different difficulty proxies (Sec. 6.2). Experimental details are in Appx. E.1.

### 6.1 MAIN RESULTS

In this work, we primarily experiment on verifiable tasks, including MATH500, GSM8K, and LiveCodeBench, and a non-verifiable task, GPQA-Diamond (Rein et al., 2024), with models Qwen2.5-

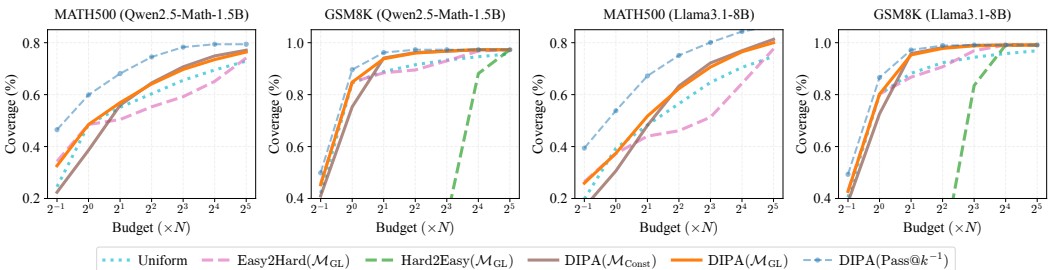

Figure 1: Performance comparison of `DIPA` against other allocation strategies across two datasets and models. The (invisible) coverages of Hard2Easy($\mathcal{M}_{GL}$) are always less than 0.2 on MATH500.

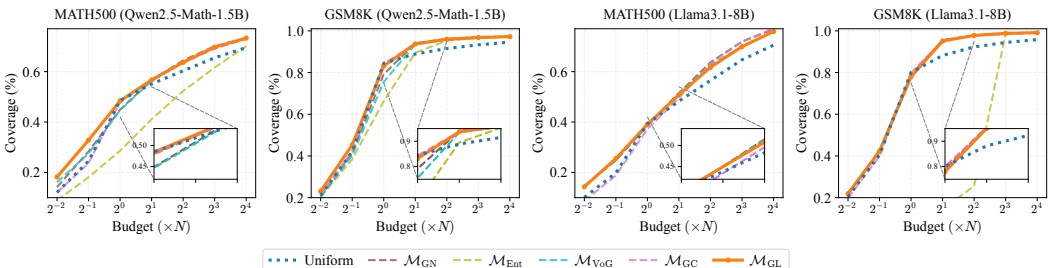

Figure 2: Performance comparison of `DIPA` variants with different proxies across models and datasets.

Math-1.5B and Llama3.1-8B. Our experiments are designed to validate the core claims regarding the ability of `DIPA` to leverage difficulty proxies for superior compute allocation.

**Compelling Performance of Probabilistic Allocation.** We first compare `DIPA` against three deterministic allocation strategies: (1) **Uniform**, which distributes $T$ equally; (2) **Easy2Hard**, which greedily selects the instance with the current lowest estimated difficulty; (3) **Hard2Easy**, which conversely selects the instance with the highest estimated difficulty. For `DIPA`, we evaluate three key variants: `DIPA`($\mathcal{M}_{Const}$) using a uniformly random sampling with a fixed constant difficulty proxy (effectively ablating the difficulty guidance), our main proposal `DIPA`($\mathcal{M}_{GL}$) guided by Generation Length, our consistently performant proxy shown in Tab. 1, and `DIPA`(Pass@$k^{-1}$) with the oracle difficulty Pass@$k^{-1}(\tau)$ defined in Eq. 3, the desired solution we aim to approximate. Experiments are conducted across varying total budgets (e.g., $T$ up to $2^5 \times N$, where $N$ is the number of instances).

The compelling results in Fig. 1 show that `DIPA` consistently outperforms or matches other methods. Specifically, `DIPA` is significantly better than uniform allocation, and a lot more as the budget increases. When compared to deterministic strategies, `DIPA`($\mathcal{M}_{GL}$) exhibits similar coverage to Easy2Hard($\mathcal{M}_{GL}$) at very small budgets, as both prioritize apparently easy instances. However, as the budget is increasing ($T \geq 2^0 \times N$), `DIPA`($\mathcal{M}_{GL}$) pulls ahead significantly. This superiority underscores the benefit of the probabilistic nature and dynamic difficulty updates of `DIPA`. Unlike the greedy Easy2Hard, `DIPA` can strategically explore instances initially deemed harder or re-allocate resources if initial difficulty estimates prove inaccurate, aligning with the exploration-exploitation balance inherent in our MAB formulation. This adaptability allows `DIPA` to solve a broader and more challenging set of problems as more compute becomes available. Crucially, `DIPA`($\mathcal{M}_{GL}$) significantly surpasses `DIPA`($\mathcal{M}_{Const}$) across most budget regimes, particularly when $T \leq 2^1 \times N$. This empirically substantiates our claim that leveraging difficulty information (e.g., $\mathcal{M}_{GL}$ here) is vital for efficient allocation, especially under tighter budget constraints. The performance gap between `DIPA` with effective proxies and the oracle (Pass@$k^{-1}(\tau)$) is observed to be smaller on GSM8K than on MATH500, which we attribute to the comparatively higher intrinsic difficulty of MATH500 dataset for the models tested.

**GL for Math, VoG for Code.** To further validate our findings from Sec. 4.2 regarding proxy quality, we compare variants of `DIPA` guided by different difficulty proxies. The results, presented in Fig. 2, demonstrate that most (except $\mathcal{M}_{Ent}$) of the investigated training-free proxies enable `DIPA` to outperform uniform allocation. More importantly, the performance ranking of `DIPA` variants guided by these proxies generally aligns with the Spearman correlations of these proxies reported in Tab. 1. Specifically, `DIPA` guided by $\mathcal{M}_{GL}$ consistently yields the best performance for both MATH500 and GSM8K. Conversely, the results on LiveCodeBench in Appx. F.2 demonstrate that the proxy $\mathcal{M}_{VoG}$

Table 2: Accuracy comparison on non-verifiable task GPQA-Diamond with Qwen2.5-1.5B across budgets.

| Method | $2^2 \times N$ | $2^4 \times N$ | $2^6 \times N$ |
|---|---|---|---|
| Self-Consistency | 0.211 | 0.199 | 0.188 |
| Best-of-N | 0.205 | 0.219 | 0.229 |
| DIPA($\mathcal{M}_{\text{Ent}}$) | 0.134 | 0.224 | 0.237 |
| DIPA($\mathcal{M}_{\text{GN}}$) | 0.189 | 0.220 | 0.235 |
| DIPA($\mathcal{M}_{\text{VoG}}$) | 0.188 | **0.229** | 0.239 |
| DIPA($\mathcal{M}_{\text{GC}}$) | **0.222** | 0.222 | 0.237 |
| DIPA($\mathcal{M}_{\text{GL}}$) | 0.218 | 0.227 | **0.242** |

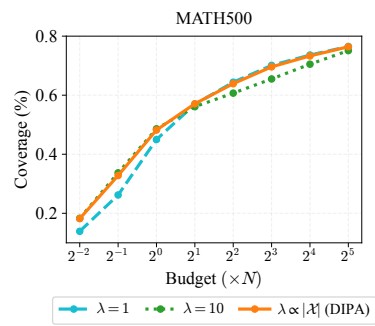

Figure 3: Comparison of different values of $\lambda$ applied to $P_k$ on MATH500.

is consistently better, which also aligns with its highest rank correlation. The relatively different performances directly confirm the significant impact of evaluating the proxy quality across domains, highlighting the importance of selecting high-fidelity difficulty proxies for practical applications.

**Generalization to Non-Verifiable Tasks.** We further validate the effectiveness of DIPA when no oracle verifier is available, where we use an off-the-shelf reward model "Skywork-Reward-V2-Qwen3-4B" (Liu et al., 2025) in the indicator function $\mathbf{1}_{\text{RM}}(\mathbf{o} \mid \mathbf{x})$. We compare DIPA against the two most common test-time strategies: Self-Consistency and Best-of-N on a challenging Q&A benchmark, GPQA-Diamond (Rein et al., 2024). To demonstrate its practicality, we report the accuracy (instead of coverage) across different budgets in Tab. 2. All variants (different proxies) of DIPA outperform both Self-Consistency and Best-of-N baselines when the budget is ample (i.e., $T \geq 2^4 \times N$), with $\mathcal{M}_{\text{GC}}$ and $\mathcal{M}_{\text{GL}}$ consistently performing better across all budgets. This demonstrates DIPA's generalization to non-verifiable domains.

**Comparison against Training-Based Difficulty Proxy Method.** Compared to the training-based baseline (Damani et al., 2025) that requires training $4.3 \times 10^6$ parameters for $54$ GPU hours, DIPA has no training overhead and achieves $50\times$ faster inference. All variants of DIPA clearly demonstrated significant advantages of efficiency and higher coverages compared to the training-based baseline. Refer to Appx. F.3 for full results.

### 6.2 ABLATION STUDIES

To dissect the contributions of the components in DIPA, we conduct several ablations to demonstrate the effectiveness and validate the principled design of DIPA. For example, we studied the role of active sampling temperature $\lambda$ in DIPA. The sampling probability $P_k$ (Eq. 5) incorporates $\lambda$ to modulate the exploration-exploitation balance. We compare fixed $\lambda$ values (1, 10) with a dynamic $\lambda \propto |\mathcal{X}|$ (where $|\mathcal{X}|$ is the count of unsolved instances). Fig. 3 shows that a dynamic $\lambda$ often performs best. It encourages exploitation early on (larger $|\mathcal{X}|$, larger $\lambda$) and shifts towards exploration as fewer, likely harder, instances remain (smaller $|\mathcal{X}|$, smaller $\lambda$). Due to the space constraint, we presented more ablations in Appx. F, including *(a)* DIPA dynamically allocates budget across difficulty levels (in Appx. F.4), *(b)* Comparison between generation-based difficulty estimates and static input estimates (in Appx. F.5), *(c)* Correlation evaluation for subdomains and analysis of non-solvable problems in MATH500 (in Appx. F.6), *(d)* How the entropy loss affects loss-based difficulty proxies, including $\mathcal{M}_{\text{Ent}}$, $\mathcal{M}_{\text{GN}}$, and $\mathcal{M}_{\text{VoG}}$ (in Appx. F.7). Those results confirmed the efficacy and design of DIPA.

## 7 CONCLUSION

We presented DIPA, a training-free approach for adaptive test-time compute allocation in LLMs. By leveraging dynamically updated difficulty proxies within a novel MAB framework, DIPA significantly enhances resource efficiency, solving more problems on challenging benchmarks under fixed compute budgets. This work offers a practical and theoretically grounded approach for cost-effective and adaptive LLM inference. Refer to Appx. A for discussions on limitations of this work.

ETHIC STATEMENT

This paper presents DIPA algorithm with the goal of advancing LLM test-time compute efficiency. Specifically, we focus on improving the efficiency of LLM test-time compute scaling by adaptive allocation, which can significantly reduce computational resource requirements during LLM inference. While there could be potential societal consequences of our work, none of which we feel must be specifically highlighted here.

REPRODUCIBILITY STATEMENT

We discuss the efforts that have been made to ensure reproducibility of our work here. We provided the source code as part of the supplemental materials. The assumption and the complete proof of our main theorem are included in Appx. D. The experimental setting and choices of hyperparameters are detailed in Appx. E.

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

APPENDIX

## A    LIMITATION

One limitation of this work is that it does not extend to the adaptive reasoning length setting, where test-time compute scaling would dynamically control the reasoning length (or the number of thinking tokens) within a single response, rather than generating multiple responses. More test-time compute means longer reasoning lengths for a single generation. In future research, we aim to extend our DIPA algorithm to the adaptive length setting for efficient thinking.

## B    EXTENDED RELATED WORKS

**Scaling LLM Test-Time Compute**.  Recent works Wu et al. (2025a); Snell et al. (2024) have demonstrated that scaling inference compute with inference strategies can be more computationally efficient than scaling model parameters. We categorize test-time compute scaling approaches into two main categories: *prompting-based* methods and *sampling-based*.  Sampling-based methods Wang et al. (2023); Chen et al. (2024b); Snell et al. (2024) generate multiple outputs simultaneously, with intermediate steps or final answers evaluated by verifiers to produce an aggregated result. The most widely adopted method is to best-of-n or beam search ranked by a verifier (or reward model). When verifiers are not available, self-consistency Wang et al. (2023) can be employed to select the consensus answer with majority vote. Prompting-based methods Wei et al. (2022); Bi et al. (2024) typically involve "thinking with more sequential tokens", where self-correction Shinn et al. (2023) is a common method that revises the initial response without external feedback. Our work falls within the sampling-based category, with our primary goal being to adaptively scale test-time compute for optimized efficiency.

**Efficient Reasoning (ER)**. Current works on efficient reasoning can be classified into three main categories Sui et al. (2025): (1) model-based ER Hao et al. (2024); Yeo et al. (2025); Zhang et al. (2025a), which optimizes the output generation length by training length-aware models, (2) output-based ER Snell et al. (2024); Wu et al. (2025b), which aims to reduce reasoning length or adaptively scale test-time compute, and (3) prompt-based ER Han et al. (2024); Ma et al. (2025), which seeks to adjust reasoning efforts by prompting models using difficulty or length control. Our work belongs to the second category, where reasoning efficiency is achieved by reducing the number of samples required during the inference process.

## C    THE USE OF LARGE LANGUAGE MODELS

We disclose that there is no significant LLM usage in this work. LLMs are only moderately used to polish writing for a few paragraphs and assist with some code implementations.

## D    PROOF OF THE REGRET BOUND FOR DIPA

This appendix establishes a regret bound for DIPA that depends only on the pairwise ranking errors between the proxy-induced order and the oracle order of arms. Intuitively, each misordered pair can contribute regret only while both arms remain active; after one fires (is solved), it stops contributing regret.

### D.1    SETUP, NOTATION, AND INVERSION COUNT

Let $\mathcal{A} = \{1, \ldots, N\}$ index arms, and let the (unknown) success probabilities be $p_1, \ldots, p_N \in [0, 1]$. Without loss of generality, label arms by decreasing oracle probabilities:

$$p_1 \geq p_2 \geq \cdots \geq p_N.$$

Let the static proxy induce desirability weights $w_i > 0$ and define the proxy order $\sigma$ such that

$$w_{\sigma(1)} \geq w_{\sigma(2)} \geq \cdots \geq w_{\sigma(N)}.$$

Define the Kendall–tau inversion set

$$\mathrm{Inv} := \big\{(i,j): \ i < j \text{ and } w_i < w_j\big\},$$

with inversion count $K := |\mathrm{Inv}|$. Thus $(i,j) \in \mathrm{Inv}$ means the proxy ranks arm $j$ above $i$ despite $p_i \geq p_j$.

**Assumption 1** (Proxy quality $\gamma$). *There exists $\gamma \geq 1$ such that for all $i, j \in \mathcal{A}$,*

$$\frac{1}{\gamma} \ \leq \ \frac{(w_i/p_i)}{(w_j/p_j)} \ \leq \ \gamma.$$

*Equivalently, $\frac{w_i}{w_j} \in \big[\frac{1}{\gamma} \cdot \frac{p_i}{p_j}, \ \gamma \cdot \frac{p_i}{p_j}\big]$.*

At round $t$, let $\mathcal{A}_t$ be the active set, and DIPA selects arm $k \in \mathcal{A}_t$ with probability

$$P_k(t) \ = \ \frac{w_k}{\sum_{u \in \mathcal{A}_t} w_u}.$$

The single-step expected regret is

$$\Delta_t \ = \ \max_{k \in \mathcal{A}_t} p_k \ - \ \sum_{k \in \mathcal{A}_t} P_k(t)\, p_k.$$

We study the cumulative regret $R(T) = \sum_{t=1}^{T} \mathbb{E}[\Delta_t]$.

### D.2  PAIRWISE CONTRIBUTION DECOMPOSITION

Fix a time $t$ and let $i^\star \in \mathcal{A}_t$ be the best active arm, i.e., $p_{i^\star} = \max_{k \in \mathcal{A}_t} p_k$. Since we have

$$
\begin{aligned}
\sum_k P_k(t)\, p_k &= p_{i^\star}\, P_{i^\star}(t) \ + \ \sum_{j \neq i^\star} P_j(t)\, p_j \\
&= p_{i^\star} \Big(1 - \sum_{j \neq i^\star} P_j(t)\Big) \ + \ \sum_{j \neq i^\star} P_j(t)\, p_j \\
&= p_{i^\star} \ - \ \sum_{j \neq i^\star} (p_{i^\star} - p_j)\, P_j(t).
\end{aligned}
$$

Then

$$\Delta_t \ = \ \sum_{\substack{j \in \mathcal{A}_t \\ j \neq i^\star}} (p_{i^\star} - p_j)\, P_j(t). \tag{6}$$

The cumulative regret therefore is

$$R(T) \ = \ \sum_{t=1}^{T} \sum_{\substack{j \in \mathcal{A}_t \\ j \neq i^\star(t)}} (p_{i^\star(t)} - p_j)\, \mathbb{E}\big[P_j(t)\big], \tag{7}$$

where $i^\star(t)$ denotes the best active arm at time $t$.

We will control the contribution in equation 7 by summing over misordered pairs $(i,j) \in \mathrm{Inv}$ and bounding how much total probability mass DIPA allocates to $j$ while both $i$ and $j$ are active.

### D.3 TWO-ARM ABSORPTION BOUND FOR A MISORDERED PAIR

Fix an inverted pair $(i, j) \in \mathrm{Inv}$ (so $i < j$ and $p_i \geq p_j$ but $w_i < w_j$). Consider the stochastic sub-process that evolves only until the earlier of: arm $i$ succeeds or arm $j$ succeeds. During rounds when both $i$ and $j$ are active, define

$$\alpha_t := P_i(t) = \frac{w_i}{\sum_{u \in \mathcal{A}_t} w_u}, \qquad \beta_t := P_j(t) = \frac{w_j}{\sum_{u \in \mathcal{A}_t} w_u}.$$

Let $\tau_{i,j}$ be the (random) absorption time when either $i$ or $j$ is solved. The total probability mass allocated to $j$ while the pair is active is $\sum_{t=1}^{\tau_{i,j}-1} \beta_t$. Define the pairwise contribution to cumulative regret (dominated by placing mass on $j$ rather than on the best active arm) as

$$\mathcal{C}_{i,j} := \sum_{t=1}^{T} \mathbb{E}\big[(p_i - p_j)\, P_j(t)\, \mathbf{1}\{i, j \in \mathcal{A}_t\}\big].$$

Now, since the sum over $t$ stops once either $i$ or $j$ is solved, we can replace the upper limit $T$ by $\tau_{i,j}$:

$$\mathcal{C}_{i,j} = (p_i - p_j) \cdot \mathbb{E}\Big[\sum_{t=1}^{\tau_{i,j}-1} \beta_t\Big]. \tag{8}$$

We next upper-bound $\mathbb{E}[\sum_{t<\tau_{i,j}} \beta_t]$ using Assumption 1. For rounds $t < \tau_{i,j}$,

$$\frac{\beta_t}{\alpha_t} = \frac{w_j}{w_i} \leq \gamma \frac{p_j}{p_i}, \qquad \frac{\alpha_t}{\beta_t} \geq \frac{1}{\gamma} \frac{p_i}{p_j}.$$

Moreover, the per-round absorption probability (either $i$ or $j$ succeeds and is removed) via $\{i, j\}$ is at least

$$\nu_t := \alpha_t\, p_i + \beta_t\, p_j \geq \alpha_t\, p_i.$$

Consider a non-homogeneous geometric process: in a sequence of Bernoulli trials, the success probability can vary with time: at round $t$, success occurs with probability $\nu_t$. The stopping time $\tau$ is the first round where success occurs. Formally, the success probability occurring at time $t$ is:

$$\Pr(\tau = t) = \nu_t \prod_{s=1}^{t-1} (1 - \nu_s)$$

At each round $t$, the probability that the process survives until $t$ (i.e., no success yet) is $\prod_{s=1}^{t-1}(1 - \nu_s)$. So,

$$\mathbb{E}\left[\sum_{t=1}^{\tau} \beta_t\right] = \sum_{t=1}^{\infty} \beta_t \prod_{s=1}^{t-1}(1 - \nu_s) = \sum_{t=1}^{\infty} \frac{\beta_t}{\nu_t} \nu_t \prod_{s=1}^{t-1}(1 - \nu_s) = \sum_{t=1}^{\infty} \frac{\beta_t}{\nu_t} \Pr(\tau = t).$$

This is a convex combination of the ratios $\beta_t/\alpha_t$.

Therefore,

$$\mathbb{E}\Big[\sum_{t=1}^{\tau_{i,j}-1} \beta_t\Big] \leq \sup_{t<\tau_{i,j}} \frac{\beta_t}{\nu_t}.$$

Combining all inequalities above, we have

$$\mathbb{E}\Big[\sum_{t=1}^{\tau_{i,j}-1} \beta_t\Big] \leq \sup_{t<\tau_{i,j}} \frac{\beta_t}{\nu_t} \leq \sup_{t<\tau_{i,j}} \frac{\beta_t}{\alpha_t\, p_i} = \frac{1}{p_i} \cdot \sup_{t<\tau_{i,j}} \frac{\beta_t}{\alpha_t} \leq \frac{\gamma}{p_i} \cdot \frac{p_j}{p_i} = \gamma \frac{p_j}{p_i^2}. \tag{9}$$

The first inequality uses the renewal theory that we can upper-bound the cumulative mass on $j$ by the largest possible ratio $\beta_t/a_t$ over the active period, by leveraging the convex combination of $\beta_t/a_t$; the second inequality uses $a_t \geq \alpha_t p_i$; the last uses Assumption 1.

Combining equation 8 and equation 9 yields

$$\mathcal{C}_{i,j} \ \leq \ (p_i - p_j) \cdot \gamma \, \frac{p_j}{p_i^2} \ \leq \ \gamma \, \frac{(p_i - p_j)}{p_i}, \tag{10}$$

since $p_j \leq p_i$. In particular, $\mathcal{C}_{i,j} \leq \gamma$ for all $(i,j) \in \mathrm{Inv}$ because $p_i \in (0,1]$.

### D.4 MAIN THEOREM: T-INDEPENDENT INVERSION BOUND

We now aggregate the pairwise contributions across inversions.

**Theorem 2** (simplified). *Under Assumption 1 and with static weights $w$, the cumulative regret of DIPA after $T$ rounds satisfies*

$$R(T) \ \leq \ \sum_{(i,j) \in \mathrm{Inv}} \gamma \, \frac{(p_i - p_j)}{p_i} \ \leq \ \gamma \, K.$$

*where $K = |\mathrm{Inv}|$. In particular, $R(T)$ is independent of $T$ and scales at most linearly with the inversion count.*

*Proof.* At each round $t$, decompose the instantaneous regret via equation 6. Partition the sum over $j$ into two classes: those $j$ that form an inversion with the best active arm $i^\star(t)$, and those that do not. The non-inverted pairs $(i^\star(t), j)$ have $w_{i^\star(t)} \geq w_j$, hence (heuristically) place less probability mass on $j$. The contribution from non-inverted pairs is always less than or equal to what would arise if the pair were inverted. Therefore, for an upper bound, it suffices to sum over inverted pairs only.

For an inverted pair $(i,j) \in \mathrm{Inv}$, its contribution to the regret persists only while both arms are active; once either is eliminated, $(i,j)$ stops to contribute. Summing over time and applying equation 10, we have

$$\sum_{t=1}^{T} \mathbb{E}\big[(p_{i^\star(t)} - p_j)\, P_j(t) \cdot \mathbf{1}\{i, j \in \mathcal{A}_t\}\big] \ \leq \ \gamma \, \frac{(p_i - p_j)}{p_i},$$

where we used $i^\star(t) \in \{i, j\}$ while both are active, and $p_{i^\star(t)} \leq p_i$ for $(i,j) \in \mathrm{Inv}$ by oracle ordering. Summing over all $(i,j) \in \mathrm{Inv}$ gives the first inequality in Thm. 2. The second inequality follows by noting $(p_i - p_j)/p_i \leq 1$ and summing $K$ terms. $\square$

### D.5 REMARKS

To simplify the proof, we use the static weights for proxies. When weights are dynamically updated over time, the same argument applies piecewise between update epochs; if updates reduce inversions monotonically in expectation, the bound remains $T$-independent and improves with calibration. Empirically, we also verified in Appx. F.5 that the dynamically updated weight has consistently higher coverages than its static invariant, indicating achieving a lower regret.

## E EXPERIMENTAL DETAILS

### E.1 EXPERIMENT SETTING

**Models and Datasets.** We primarily experiment with three LLMs including Qwen2.5-Math-1.5B-Instruct Yang et al. (2024), Qwen2.5-1.5B-Instruct Yang et al. (2024), and Llama3.1-8B Grattafiori et al. (2024). To conduct experiments on a verifiable domain (e.g., mathematical reasoning), we use the official test split from MATH500 and GSM8K, with an oracle verifier to check if the answer is correct.

**Evaluation.** Since both the generation process is stochastic and our DIPA method employs probabilistic sampling, we conducted three independent runs using 3 random seeds for our experiments. The reported coverage or accuracy represents the average across these three runs, rounded to the third decimal place.

**Hyperparameters.** For each question from the test dataset, we randomly sampled 500 responses independently by applying the prompt template in Appx. E.2, where zero-shot evaluation is used. For sampling parameters, we set temperature to 0.7, top_p to 0.8, repetition_penalty to 1.05, and max_tokens to 512. For calculating the correlation in Tab. 1, if there is no any attempt is correct within the 500 responses, we manually set its $\text{Pass@}k^{-1}(\tau) = 1000$. In practice, due to the precision of floating number, we set $\tau = 0.99$.

**Computational Resources.** All experiments are conducted on a server of NVIDIA 8×H100 PCIe (81559MB). Sampling 500 responses from each dataset (i.e., MATH500 and GSM8K) typically requires less than 2 GPU hours, and calculating difficulty proxies for all responses usually require less than 48 GPU hours.

### E.2 PROMPT TEMPLATE

> **Prompt Template for Mathematical Reasoning**
>
> **System: Please reason step by step, and put your final answer within \\boxed{}.**
>
> **User: [Problem Content]**

### E.3 DETAILS OF DIFFICULTY PROXIES

We formally define how the entropy loss is calculated here. Note that $S$ is the sequence length of the input instance and $L_i$ is the sequence length of the corresponding generation.

$$\text{CE}(\mathbf{x} \oplus \mathbf{o}) = \frac{1}{S + L_i - 1} \left( -\sum_{s=1}^{S} \log p(x_s | \mathbf{x}_{\prec s}) - \sum_{l=1}^{L_i - 1} \log p(o_l | \mathbf{x}, \mathbf{o}_{\prec l}) \right) \tag{11}$$

For input-based proxies, we calculate the cross-entropy loss on the input tokens, instead of the whole sequence (i.e., input tokens + generation tokens). We define the input-based cross-entropy loss on the input tokens $\mathbf{x}$ with next token prediction as:

$$\text{CE}(\mathbf{x}) = \frac{1}{S - 1} \left( -\sum_{s=1}^{S-1} \log p(x_s | \mathbf{x}_{\prec s}) \right)$$

We formally introduce several training-free difficulty proxies for LLMs:

| | | |
|---|---|---|
| **Entropy:** | $\mathcal{M}_{\text{Ent}}(\mathbf{x}) \triangleq \text{CE}(\mathbf{x}) \ .$ | (12) |
| **Gradient Norm:** | $\mathcal{M}_{\text{GN}}(\mathbf{x}) \triangleq \text{Mean}\left\{\|\nabla_x \text{CE}(\mathbf{x})\| \mid x \in \mathbf{x}\right\} \ .$ | (13) |
| **Variance of Gradient:** | $\mathcal{M}_{\text{VoG}}(\mathbf{x}) \triangleq \text{Var}\left\{\|\nabla_x \text{CE}(\mathbf{x})\| \mid x \in \mathbf{x}\right\} \ .$ | (14) |

We also provide the ablation to justify our choice of calculating the loss over the entire sequence length in Appx. F.7.

**Implementation of Proxies with Negative Correlation** As we prioritize solving easier questions, we place higher probability mass on them by applying Eq. 5. For proxies that have a negative correlation (e.g., $\mathcal{M}_{\text{Ent}}$) with the oracle difficulty, we use its inverse in Eq. 5 (e.g., $1/\mathcal{M}_{\text{Ent}}$).

# F ADDITIONAL RESULTS

## F.1 RESULTS ON CORRELATION EVALUATION

We provide additional empirical results to demonstrate the valid correlation of the selected proxies. Among all evaluated metrics, Generation Length consistently emerges as a particularly compelling proxy due to its simplicity and strong empirical performance. This is visually underscored in Fig. 4, which reveals a clear positive relationship between Generation Length and the oracle difficulty, reinforcing the intuition that more complex problems often necessitate longer reasoning chains. The relationship between the sample size for estimating the correlation and the correlation value is shown in Fig. 5, where we see the marginal gains diminish rapidly, confirming the practical usage of difficulty estimation using those proxies.

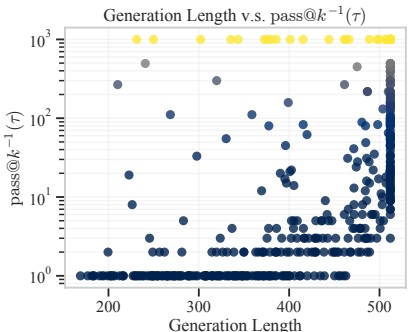

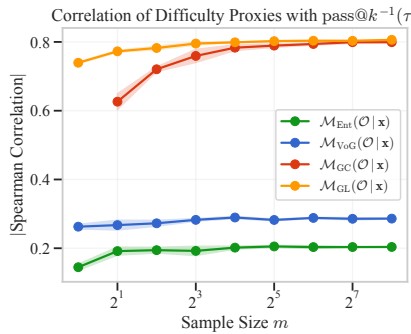

Figure 4: Scatter plot of Generation Length for instances in MATH500 and their oracle difficulties.

Figure 5: Comparison of correlations between difficulty proxies and $\text{Pass}@k^{-1}(\tau)$ across different sample size $m$ on MATH500.

## F.2 RESULTS ON LIVECODEBENCH

We extend our evaluation of DIPA to a popular code generation task LiveCodeBench (Jain et al., 2025). The results in Tab. 3 show that the proxy $\mathcal{M}_{\text{VoG}}$ performs the best among all selected proxies, aligning with its highest rank correlation reported in Tab. 1.

Table 3: Coverage Comparison on LiveCodeBench with Llama3.1-8B

| Allocation | $T = 2^{-1} \times N$ | $T = 2^1 \times N$ | $T = 2^3 \times N$ | $T = 2^5 \times N$ |
|---|---|---|---|---|
| Uniform | 0.103 | 0.265 | 0.375 | 0.474 |
| DIPA($\mathcal{M}_{\text{GN}}$) | 0.145 | 0.274 | 0.371 | 0.503 |
| DIPA($\mathcal{M}_{\text{Ent}}$) | 0.063 | 0.181 | 0.341 | 0.484 |
| DIPA($\mathcal{M}_{\text{VoG}}$) | **0.150** | **0.278** | **0.377** | **0.518** |
| DIPA($\mathcal{M}_{\text{GC}}$) | 0.101 | 0.270 | 0.366 | 0.501 |
| DIPA($\mathcal{M}_{\text{GL}}$) | 0.133 | **0.278** | 0.373 | 0.507 |

## F.3 COMPARISON WITH TRAINING-BASED METHOD

To highlight the efficiency of the proposed training-free approach, we compare DIPA with the recent difficulty-adaptive inference method from Damani et al. (2025), which uses training-based difficulty estimation. As the implementation of Damani et al. (2025) is not publicly available, we have implemented the method from Damani et al. (2025) using parameter-efficient fine-tuning (LoRA) on Qwen2.5-1.5B base LM, training it on 12k MATH training data as specified in their work for the math setting. Both our method and Damani et al. (2025) (at inference) are evaluated on MATH500. We report the comparison of coverage in Tab. 4 and that of computational resources in Tab. 5.

Table 4: Comparison of coverages against the training-based baseline on MATH500 with Qwen2.5-Math-1.5B

| Method | $T = 2^{-1} \times N$ | $T = 2^1 \times N$ | $T = 2^3 \times N$ | $T = 2^5 \times N$ |
|---|---|---|---|---|
| Damani et al. (2025) | 0.0 | 0.409 | 0.594 | 0.677 |
| DIPA($\mathcal{M}_{\text{VoG}}$) | 0.281 | 0.540 | 0.677 | 0.758 |
| DIPA($\mathcal{M}_{\text{GC}}$) | 0.241 | **0.573** | 0.689 | 0.759 |
| DIPA($\mathcal{M}_{\text{GL}}$) | **0.333** | 0.565 | **0.692** | **0.765** |

Table 5: Comparison against baseline on GPU hours, training parameters, and inference time on MATH500 with base model Qwen2.5-1.5B as the proxy in Damani et al. (2025).

| Method | # Trainable Param. | Training (h) | Inference (s) |
|---|---|---|---|
| Damani et al. (2025) | $4.3 \times 10^6$ | 53.93 | 105.75 |
| DIPA($\mathcal{M}_{\text{VoG}}$) | 0 | 0 | 1.97 |
| DIPA($\mathcal{M}_{\text{GC}}$) | 0 | 0 | 0.81 |
| DIPA($\mathcal{M}_{\text{GL}}$) | 0 | 0 | 1.62 |

### F.4 DYNAMIC BUDGET ALLOCATION ACROSS DIFFICULTY LEVELS

Fig. 6 shows the adaptive budget distribution of DIPA. With small total budgets $T$, DIPA prioritizes easier instances (e.g., Levels 1-2) for quick wins. As $T$ increases, it shifts resources to harder instances (e.g., Levels 4-5), reflecting an optimal strategy to maximize solved problems. This dynamic reallocation, driven by instance completions and updated difficulty estimates, is a key strength.

### F.5 IMPORTANCE OF DYNAMIC, GENERATION-BASED DIFFICULTY UPDATES

We ablate the difficulty update mechanism (Algo. 1, Line 13) by comparing: (1) DIPA with only initial input-based proxy $\mathcal{M}_{\text{init}}$, (2) DIPA with generation-based proxy $\mathcal{M}_{\text{gen}}$ (evaluated using first $m$ attempts, then static), and (3) DIPA (our full method with continuous update). Fig. 7 shows that our full method, i.e., $\mathcal{M}_{\text{init}}$&$\mathcal{M}_{\text{gen}}$(DIPA), outperforms its static variants especially in the low-budget regime. This confirms that generation-based proxies provide more refined signals with interaction, and continuous updates allow DIPA to correct initial misjudgments and adapt its allocation effectively.

### F.6 FURTHER ANALYSIS ON MATH500

**Problem Types.** As math problems contain different subdomains (i.e., problem types), we further analyze the Spearman correlation of different proxies under varying problem types. The result shown in Tab. 6 indicates that proxy $\mathcal{M}_{\text{GL}}$ (i.e., Generation Length) consistently performs well across all

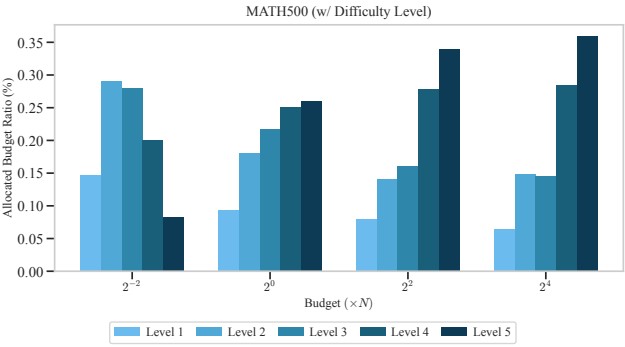

Figure 6: Allocation outcome of DIPA on MATH500 varying difficulty levels. The allocated budget ratio is calculated based on the budget spent on each difficulty level across different budgets.

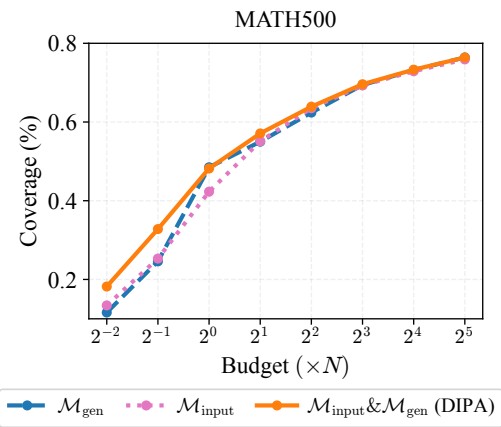

Figure 7: Comparison of difficulty update mechanisms on MATH500.

problem types and has a lowest variance of correlations, which suggests its robustness and generality on different subdomains.

Table 6: Correlation comparisons of different proxies separated by problem type on MATH500.

| Type | $\mathcal{M}_{GL}$ | $\mathcal{M}_{GC}$ | $\mathcal{M}_{Ent}$ | $\mathcal{M}_{VoG}$ |
|---|---|---|---|---|
| Algebra | 0.818 | -0.760 | -0.099 | -0.317 |
| Counting & Probability | 0.667 | -0.630 | -0.084 | -0.158 |
| Geometry | 0.746 | -0.573 | -0.477 | -0.005 |
| Intermediate Algebra | 0.561 | -0.467 | -0.182 | -0.064 |
| Number Theory | 0.803 | -0.675 | 0.103 | -0.269 |
| Prealgebra | 0.663 | -0.670 | -0.352 | 0.014 |
| Precalculus | 0.616 | -0.271 | 0.157 | -0.366 |
| Variance | **0.009** | 0.027 | 0.052 | 0.024 |

**Non-Solvable Problems.** Although MATH500 has human-annotated difficulty level (i.e., Level 1-5), we further study its oracle difficulty (based on Pass@$k^{-1}(\tau)$) by categorizing the problems using Pass@$k^{-1}(\tau)$, where Pass@$k^{-1}(\tau) = 1$ represents easy problems that can be solved using a single attempt, Pass@$k^{-1}(\tau) > 500$ represents potentially non-solvable problems within 500 attempts, and $1 < $ Pass@$k^{-1}(\tau) \leq 500$ represent problems that are solvable with varying difficulty. The distribution of the three categories of problems is shown in Fig. 8, where the easy problems (i.e, Pass@$k^{-1}(\tau) = 1$) have 28% problem instances of the whole test set. This suggests that an allocation algorithm should uses a moderate exploration (or non-zero probability) that at least attempt every problem once if the budget allows to cover easy problems.

In addition, the relative large proportion (i.e., 20%) of non-solvable problems with Pass@$k^{-1}(\tau) > 500$ in Fig. 8 indicates that the efficiency of an allocation algorithm can be further improved if those non-solvable problems could be effectively identified (potentially through our proposed difficulty proxies) to avoid non-necessary budget cost.

### F.7 SEQUENCE LOSS OR GENERATION LOSS

We further study the difficulty proxies that are associated with the entropy loss (or its back-propagated gradients). To justify our choice of calculating the entropy loss on the entire sequence (i.e., CE($\mathbf{x} \oplus \mathbf{o}$)), we compare the correlations produced by those proxies with entropy loss only on the generation tokens (i.e., CE($\mathbf{o}$)). The result in Fig. 9 shows that the correlations are consistently higher when using the entropy loss calculated on the entire sequence, which confirms our choice.

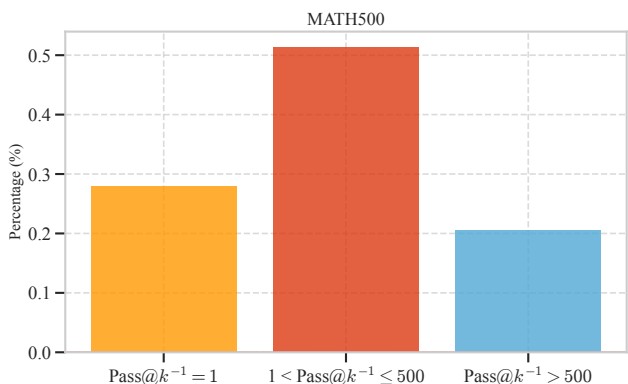

Figure 8: Percentage of problems in MATH500 that are categorized into three difficulty levels.

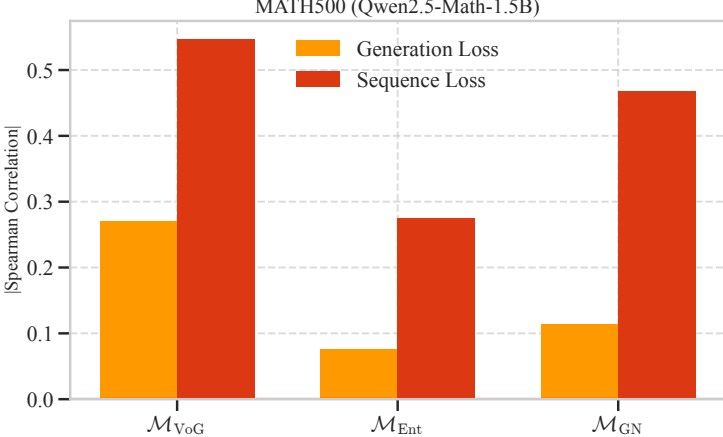

Figure 9: Correlation comparison across different proxies using different loss calculation methods.

### F.8 RESULTS ON OTHER MODELS

In addition to the results on Qwen2.5-Math-1.5B and Llama3.1-8B provided in Fig. 1 and 2, we also conduct similar experiments using the general model (not math finetuned) Qwen2.5-1.5B. As shown in Fig. 10, we find DIPA($\mathcal{M}_{\mathrm{GL}}$) consistently achieves the best performance against other baseline allocation strategies on both MATH500 and GSM8K, demonstrating a comparable performance with DIPA using the oracle difficulty proxy Pass@$k^{-1}$. We attribute the larger gap between DIPA($\mathcal{M}_{\mathrm{GL}}$) and DIPA(Pass@$k^{-1}$) on MATH500 to its greater problem difficulties.

The result in Fig. 11 demonstrates the performance comparison of DIPA using different proxies, where we find most of the proxies achieve higher coverage than the uniform allocation strategy and $\mathcal{M}_{\mathrm{GL}}$ (Generation Length) being the most robust and effective one.

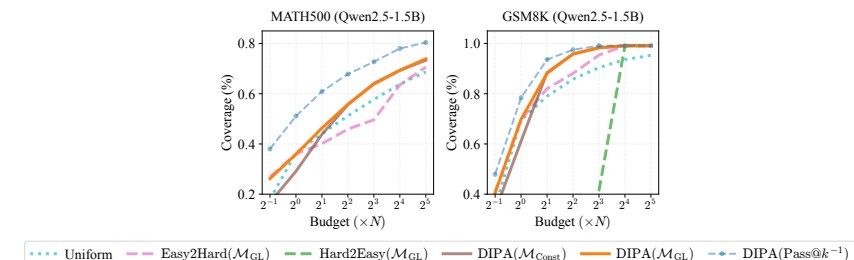

Figure 10: Performance comparison of `DIPA` against other allocation strategies across two datasets with Qwen2.5-1.5B. The (invisiable) coverages of Hard2Easy($\mathcal{M}_{\text{GL}}$) are always less than 0.1 on MATH500.

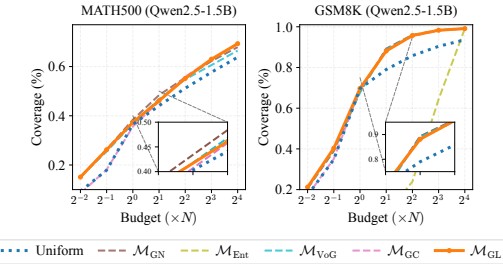

Figure 11: Performance comparison of `DIPA` variants with different proxies on MATH500 and GSM8K with Qwen2.5-1.5B.

