# OpenReview forum: "Adaptive Test-Time Compute Allocation via Training-Free Difficulty Proxies"
_ICLR.cc/2026/Conference — Submitted to ICLR 2026_

### Official Review · Reviewer_MUor · 2025-10-29

**Soundness:** 2
**Presentation:** 2
**Contribution:** 2
**Rating:** 4
**Confidence:** 4

**Summary:**

This paper tackles the problem of adaptive test-time compute allocation for large language models (LLMs). Traditional inference strategies, such as self-consistency and Best-of-N sampling, apply uniform compute budgets to all inputs, regardless of their difficulty. The authors propose DIPA, a training-free framework that dynamically allocates compute based on difficulty proxies derived directly from the LLM’s generation process (e.g., entropy, variance of gradient norms, generation length, and consistency). The method formulates adaptive allocation as a multi-armed bandit (MAB) problem, introducing probabilistic sampling to balance exploration and exploitation. Theoretically, they provide a regret bound showing that performance depends on the correlation between the proxy and true difficulty. Experiments on reasoning and coding benchmarks demonstrate that DIPA significantly outperforms uniform and training-based baselines under fixed compute budgets.

**Strengths:**

- The proposed solution is interesting. The reformulation of compute allocation as a multi-armed bandit with arm elimination is well-justified.
- The paper systematically investigates multiple training-free proxies (entropy, gradient norms, generation length, etc.) and analyzes their correlation with oracle difficult
- The paper is easy to follow.

**Weaknesses:**

- The individual proxies (e.g., generation length, entropy) are adapted from prior uncertainty estimation works. The main contribution lies in combining and evaluating them.
- The method's performance heavily depends on the proxy's correlation with true difficulty.
- Some works on test-time scaling are missing, such as [1,2,3].

[1] Inference Scaling Laws: An Empirical Analysis of Compute-Optimal Inference for LLM Problem-Solving.

[2] Inference Scaling fLaws: The Limits of LLM Resampling with Imperfect Verifiers.

[3] Can 1B LLM Surpass 405B LLM? Rethinking Compute-Optimal Test-Time Scaling.

**Questions:**

- Can the authors provide some guidelines for how to use these proxies in practical scenarios? For example, how to achieve good performance when encountering a brand new dataset?

---

> ### Author Response · Authors · 2025-12-03
> **Thank you for your review**
>
> We thank the reviewer for finding the paper easy to follow and the solution interesting. Below, we address the raised concerns.
>
> > Response to W1
>
> We agree that the individual signals (e.g., entropy) are established. However, the core novelty of DIPA is not inventing training-free proxies themselves, but our new formulation of test-time compute allocation as a Multi-Armed Bandit (MAB) problem and the novel DIPA algorithm that solves it. To clarify, DIPA's distinct novelty lies in three key aspects:
>
> 1. Training-free difficulty proxies: We empirically demonstrate that problem difficulty can be estimated through different proxies without requiring additional training (Sec. 3.1), whereas prior work relies on a learned difficulty estimator.
> 2. New MAB formulation: We reformulate our problem as a new MAB variant tailored for compute allocation.
> Difficulty-informed probabilistic allocation: We propose a novel algorithm that adaptively selects problems to solve based on dynamically updated proxy measures.
> 3. While our work and previous literature may share similar training-free proxies, DIPA's technical approach and empirical investigation are fundamentally different, establishing its novelty, especially for LLM test-time compute allocation.
>
>
> We will revise the manuscript to make our positioning and contribution clearer.
>
>
> > Response to W2
>
> We clarify that the dependence on the difficulty proxies is a design feature (allocating compute via training-free difficulty proxies), not a weakness. We have empirically demonstrated that those proxies are effective across datasets and models, establishing their robustness and practicality.
>
> > Response to W3
>
> Thank you for pointing out the missing references. We will cite and discuss them in the revision.
>
>
> > Response to Q1
>
> Based on our ablation studies (Table 1 & Fig 2), we propose the following heuristic guidelines for practitioners:
>
> * **For Reasoning/Math:** Use **Generation Length ($\mathcal{M}_{GL}$)**. It is the most robust proxy because complex reasoning usually correlates with a longer chain-of-thought.
> * **For Coding:** Use **Variance of Gradients ($\mathcal{M}_{VoG}$)**. Code generation sensitivity to input perturbations is a strong signal of difficulty.
> * **General Rule:** Start with $\mathcal{M}_{GL}$ as it requires zero additional forward passes and offers the best efficiency-performance trade-off.
>
> ---
>
> We sincerely thank the reviewer for the valuable feedback. We hope our clarifications effectively address your concerns and increase your opinion of our work.

---

### Official Review · Reviewer_cYP5 · 2025-10-30

**Soundness:** 2
**Presentation:** 2
**Contribution:** 2
**Rating:** 2
**Confidence:** 2

**Summary:**

The paper focuses on adaptive test-time compute allocation for LLM reasoning under a fixed budget, arguing that uniform “N-samples per instance” and training-based difficulty predictors are either inefficient or costly to train/deploy. The authors instead propose training-free difficulty proxies and formulates allocation as a stochastic MAB with arm-elimination, introducing DIPA, which samples instances with probability inversely proportional to their current difficulty, initializes from cheap input-based priors, and updates online using generation-based proxies.

**Strengths:**

This paper addresses the issue that training-based difficulty predictors are inefficient or costly, provide training-free difficulty proxies insights, and present corresponding experimental results.

**Weaknesses:**

- I believe that the Easy2Hard strategy aligns with DIPA’s fundamental principle of “probabilistically prioritizing arms (instances) estimated to be easier.” The reason Easy2Hard underperforms in the experiments is likely due to inaccurate initial difficulty estimation. Did the authors compare the performance of Easy2Hard when using existing training-based difficulty predictors to estimate difficulty before allocation?
- The proposed method requires sequential rollouts, which sacrifices parallelism. Although the overall compute budget is fixed, this design increases wall-clock time. While the authors mention that DIPA could, in theory, be extended to a batched version, I think that sampling multiple instances simultaneously under the same probability distribution would likely result in over-selecting easy instances, leading to budget waste on already simple problems.

**Questions:**

See weakness

---

> ### Author Response · Authors · 2025-12-03
> **Thank you for your review**
>
> We thank the reviewer for the detailed critique. We address your concerns below:
>
> > Response to W1
>
>
> We clarify that we also adaptively update the difficulty estimate in the Easy2Hard allocation strategy, and the failure of Easy2Hard is not due to poor initial priors. If a proxy (even a good one) incorrectly flags a solvable instance as "Hard," a deterministic greedy Easy2Hard strategy will never attempt it until all "Easy" ones are exhausted. DIPA, being probabilistic, assigns a small but non-zero probability to "Hard" instances, allowing it to correct estimation errors via exploration.
>
>
> We thank the reviewer for the suggestion on comparing the performance of Easy2Hard when using existing training-based difficulty predictors. We have provided the new result in Table-R2 below. As shown in Table R1, applying Easy2Hard to a trained predictor actually **degrades** performance compared to the original baseline and significantly underperforms DIPA. This confirms that probabilistic allocation (DIPA) is superior to greedy allocation (Easy2Hard), even when the difficulty estimator is strong (via training).
>
>
> **Table-R2: Comparison of Coverages on MATH500 with Qwen2.5-Math-1.5B**
> | Allocation | $T=2^{-1}\times N$ | $T=2^{1}\times N$ | $T=2^{3}\times N$ | $T=2^{5}\times N$ |
> |-|-|-|-|-|
> | Damani et al. 2025 (original) | $0.0$ | $0.409$ |$0.594$ | $0.677$ |
> | Damani et al. 2025 (Easy2Hard) | $0.044$ | $0.316$ |$0.446$ | $0.626$ |
> | DIPA($\mathcal{M}\_{\text{VoG}}$) | $0.281$ | $0.54$ | $0.677$ | $0.758$ |
> | DIPA($\mathcal{M}\_{\text{GC}}$) | $0.241$ | $\mathbf{0.573}$ | $0.689$ | $0.759$ |
> | DIPA($\mathcal{M}\_{\text{GL}}$) | $\mathbf{0.333}$ | $0.565$ | $\mathbf{0.692}$ | $\mathbf{0.765}$ |
>
>
>
>
> > Response to W2
>
>
> We clarify that our sequential formulation of DIPA (where the difficulty estimate is updated after every single attempt) inherently prioritizes compute efficiency (maximizing solved instances) over latency. For highly latency-sensitive applications, we propose to use a batched DIPA implementation, where we process a batch of $k$ instances sampled from the current distribution $P_k$ before updating the model. We acknowledge that this batched approach is sub-optimal and may lead to allocating slightly more compute on easy instances than the fully sequential method would, which presents a trade-off between latency and efficiency. However, we argue that this practical compromise still offers significant compute gains over the non-adaptive uniform allocation baseline. The "over-selecting easy instances" concern is mitigated because the probability distribution $P_k$ is updated after every batch, dynamically shifting mass away from solved instances.
>
> We will add a discussion of this efficiency-latency trade-off and the batched DIPA implementation to the main text for clarity.
>
> ---
>
> We sincerely thank the reviewer for the feedback. We hope our responses and experiments have addressed your concerns and improved your opinion of our work.

---

### Official Review · Reviewer_uSZC · 2025-10-31

**Soundness:** 2
**Presentation:** 3
**Contribution:** 2
**Rating:** 4
**Confidence:** 4

**Summary:**

This paper addresses the inefficiency of uniform test-time compute allocation for LLMs and proposes DIPA, a training-free framework for adaptive compute distribution. It leverages training-free difficulty proxies derived from LLM inputs or generation processes to estimate instance difficulty. It reformulates the allocation problem as a multi-armed bandit task with arm elimination upon success, using a probabilistic policy to balance exploration and exploitation. Experiments on varies benchmarks show DIPA outperforms uniform allocation, deterministic strategies, and training-based baselines, solving more problems under fixed budgets.

**Strengths:**

1) The proposed method achieves effective inference-budget allocation without any additional training, outperforming baselines such as BoN and SC under the same compute budget.
2) The formalization as a multi-armed bandit and the accompanying regret-bound analysis link proxy quality to performance, providing a theoretical grounding for the approach.

**Weaknesses:**

1) The experimental baselines are limited to relatively easy problems, the authors should include AIME24, a benchmark commonly used in recent reasoning-model studies, which demands greater reasoning capability and larger budgets and would better highlight the proposed method’s incremental value.
2) The selected training-free proxies are rather trivial.
3) Although casting budget allocation as a multi-armed bandit is interesting, it is not “a novel bandit formulation” but a very classical algorithmic template; the authors should acknowledge this.
4) The computational overhead of obtaining the difficulty proxies must also be accounted for—comparing only inference budgets is insufficient, explicit end-to-end run-time comparisons with each baseline are necessary.

**Questions:**

Please refer to the "Weakness" part.

---

> ### Author Response · Authors · 2025-12-03
> **Thank you for your review**
>
> We appreciate the reviewer’s recognition of the effectiveness of our proposed method DIPA, our MAB formalization, and the theoretical grounding. We address your concerns below:
>
> > Response to W1
>
> We appreciate the reviewer's suggestion, and we present **new results** on AIME-24 in **Table-R1** below. The results reveal that DIPA variants achieve only modest improvements over the baseline, and only when the computational budget reaches $T\geq2^6\times N$. We attribute this limited performance gain to two key factors: the exceptionally great difficulty of all problems in AIME-24 and the constrained capacity of our base model (i.e., 1.5B).
>
> Our method's design principle suggests it would demonstrate significantly stronger performance on benchmarks with heterogeneous difficulty distributions, where easier problems can be solved efficiently first, thereby preserving computational resources for more challenging instances. The uniformly great difficulty of AIME-24 problems prevents this adaptive resource allocation strategy from operating effectively.
>
> While time constraints during the rebuttal period prevented us from conducting experiments with more powerful base models, we recognize its significance and commit to extending our evaluation to stronger models in the revised manuscript, which we believe will further strengthen our paper.
>
> **Table-R1: Coverage Comparison on AIME24 with Qwen2.5-Math-1.5B**
> | Allocation | $T=2^{2}\times N$ | $T=2^{4}\times N$ | $T=2^{6}\times N$ | $T=2^{8}\times N$ |
> |-|-|-|-|-|
> | Uniform | $1/30$ | $1/30$ |$2/30$ | $5/30$ |
> | DIPA($\mathcal{M}\_{\text{VoG}}$) | $1/30$ | $1/30$ |$3/30$ | $\mathbf{6/30}$ |
> | DIPA($\mathcal{M}\_{\text{GC}}$) | $1/30$ | $1/30$ |$\mathbf{3/30}$ | $\mathbf{6/30}$ |
> | DIPA($\mathcal{M}\_{\text{GL}}$) | $1/30$ | $1/30$ |$\mathbf{3/30}$ | $5/30$ |
>
> > Response to W2
>
> We clarify that we have comprehensively investigated a wide range of training-free difficulty proxies, which are practical and deployable. We respectively disagree that those proxies are rather trivial. As we have shown in Table 1 that most of them present a high correlation (e.g., >0.5) with the oracle difficulty across datasets and models. We also highlight that our proposed algorithm DIPA can be equipped with more training-free proxies.
>
>
> We clarify that the "simplicity" of the proxies is a design feature, not a limitation. Our contribution is not the invention of new metrics (like entropy), but the **discovery that these simple signals can drive an efficient MAB policy**. As shown in Table 1, these simple and deployable proxies (e.g., Generation Length) achieve high correlations ($>0.5$) with oracle difficulty. Their low computational cost is exactly what makes DIPA practical for real-time allocation, fulfilling the efficiency goal of the paper.
>
>
> > Response to W3
>
>
> We clarify that while MABs are classical, our specific formulation addresses a unique setting in Test-Time Compute:
> 1. Global Budget Constraint: Unlike standard MABs that minimize regret over an infinite horizon, we maximize coverage under a strictly fixed global budget $T$.
> 2. Arm Elimination: Standard MABs continue pulling arms. Our setting involves "success" (solving the problem), after which the arm is eliminated (Line 11, Algo 1).
> 3. Regret Analysis: Our Theorem 1 specifically bounds regret based on pairwise ranking errors in this elimination setting. We will revise the text to be more precise: we apply a specialized variant of MAB to a novel application scenario.
>
>
> > Response to W4
>
> Please refer to Table 5 in Appendix F.3, which provides the requested comparison:
> - Baseline (Damani et al.): 105.75s inference + 53.93h training.
> - DIPA: 1.62s overhead + 0h training. DIPA is orders of magnitude more efficient when considering the full lifecycle. We will highlight this end-to-end comparison in the main paper.
>
> ---
>
> Thank the reviewer for their constructive feedback. We sincerely hope our responses and experiments have addressed your concerns and improved your opinion of our work.

---

### Official Review · Reviewer_t29B · 2025-11-09

**Soundness:** 3
**Presentation:** 3
**Contribution:** 2
**Rating:** 4
**Confidence:** 4

**Summary:**

This paper investigates the problem of adaptive resource allocation during test-time scaling, proposing a training-free difficulty proxy metric and providing analysis and proofs regarding the modeling and associated regret bounds via multi-armed bandits. The effectiveness of the approach is demonstrated across mathematical, programming, and document-related problems.

**Strengths:**

- The problem studied in this paper is highly important and practical.
- The proposed method has a certain mathematical foundation, though its relevance to practice remains limited.

**Weaknesses:**

- The problem is oversimplified, particularly in terms of measuring computational budget solely based on the number of steps. More specific measurements, such as FLOPs and time, are needed.
- While the paper emphasizes the theoretical contributions of the method, further demonstration of its practical application value and significance is required.
- Evaluations of computational overhead and FLOPs for inference deployment should be provided with specific calculations.

**Questions:**

- The paper also mentions the application of reward models and suggests incorporating more procedural information for difficulty analysis. It is recommended to consider the application analysis of generative reward models, such as GenGRM:

[1] GenPRM: Scaling Test-Time Compute of Process Reward Models via Generative Reasoning

---

> ### Author Response · Authors · 2025-12-03
> **Thank you for the review**
>
> We thank the reviewer for recognizing the **importance** of the problem and the soundness of our **mathematical foundation**. We address your concerns below:
>
> ---
>
> > More specific measurements, such as FLOPs and time, are needed.
>
>
> We have provided a detailed run-time analysis in Appendix F.3 (Table 5). We highlight two key findings:
>
> 1. DIPA incurs virtually zero overhead (calculating proxies and updating policies takes milliseconds) compared to the generation time.
>
> 2. The training-based method (i.e., Damani et al., 2025) requires ~54 GPU-hours for training and distinct inference overhead. In contrast, DIPA requires 0 hours of training and is immediately deployable. Since our proxies (e.g., Generation Length) are derived directly from the generated tokens or from a single back-propagation (e.g., Variance of Gradient), the additional FLOPs are negligible compared to the forward passes required for generation. We will move these specific calculations to the main text in the revision to improve visibility.
>
> >...further demonstration of its practical application value and significance is required.
>
>
> We demonstrate practical value by consistently outperforming uniform allocation on widely used benchmarks (MATH500, LiveCodeBench, GSM8K) and the Diamond-GPQA benchmark. Crucially, DIPA achieves this without the need for auxiliary models or fine-tuning, making it a practical and efficient solution for LLM inference pipelines.
>
> > Evaluations of computational overhead and FLOPs for inference deployment should be provided with specific calculations.
>
> We have included the details of evaluations in **Appendix F.3 (Table 5)**. We clarify that the run-time in Table 5 only involves the time of evaluating proxies and performing our allocation method DIPA, which excludes the time of response generation. We will clarify the specific calculations in the revised version.
>
>
> > Comparison with Generative Reward Models (e.g., GenPRM)
>
> Thank you for this relevant reference. We will cite and discuss GenPRM in the related works. Conceptually, DIPA and GenPRM address different aspects of the pipeline: DIPA is an allocation strategy (deciding which instance to allocate compute to), whereas GenPRM acts as a verifier (evaluating the quality of a generation). DIPA is orthogonal to GenPRM; one could integrate GenPRM as the verifier within the DIPA framework to further enhance performance. We believe exploring this synergy is a promising direction for future work.
>
> ---
>
> Thank you for the constructive review and valuable suggestions. We hope our responses have effectively addressed your concerns and improved your evaluation of our work.

---

### Meta-Review · Area_Chair_Wtka · 2026-01-13

**Summary:**

Across reviewers, the main concerns center on *incremental novelty* and *unclear practical impact* beyond showing that simple, training-free signals can guide adaptive allocation. Multiple reviewers note that the “difficulty proxies” (e.g., generation length / entropy / gradient-based signals) are largely standard uncertainty-like heuristics, and that the bandit framing/algorithmic template is not fundamentally new, so the contribution is perceived more as a combination + empirical study than a distinctly novel method.

On evaluation, reviewers asked for stronger/harder settings (e.g., AIME24) and more complete *end-to-end* accounting of compute and wall-clock implications (overhead of proxy computation, and the sequential nature of rollouts potentially sacrificing parallelism).

While the paper’s goal is practical (adaptive test-time compute under a fixed budget), the rebuttal’s added evidence suggests the gains can be modest on very hard / uniformly difficult benchmarks (AIME24), and the strongest practical selling point becomes “no training cost + low overhead” rather than consistently strong coverage improvements across regimes.

Overall, the method is sensible and could be useful, but based on the reviews and rebuttal, the perceived novelty is limited, and the strongest new experiments (AIME24) do not yet demonstrate robust, compelling improvements in the regimes most aligned with “reasoning-model” test-time scaling claims.

**Reviewer Concerns:**

### Addressed (at least partially)
- **Compute / overhead accounting:** Authors added runtime/training comparisons and clarified that DIPA has small overhead and avoids training, with a concrete table comparing training hours and inference-time overhead vs. a training-based baseline.
- **Harder benchmark request (AIME24):** Authors added AIME24 results and explicitly acknowledged that improvements are modest with their chosen small base model, and argue heterogeneity of difficulty is important for their method to shine.
- **Easy2Hard comparison with a trained predictor:** In response to the concern that Easy2Hard may fail due to poor priors, authors reported a new comparison showing Easy2Hard can underperform even with a trained predictor, arguing probabilistic exploration is the key differentiator.
- **Missing related work on test-time scaling:** Authors committed to adding missing citations raised by a reviewer.
- **Practical “which proxy to use” guidance:** Authors provided heuristic guidance (e.g., generation length for math/reasoning; variance-of-gradients for coding).

### Still outstanding / not fully resolved
- **Novelty / positioning remains weak:** Even after rebuttal, reviewers’ core point stands: the proxies are largely standard and the bandit machinery is close to classical templates. Authors propose rephrasing claims, but this is largely a *positioning fix* rather than new technical novelty.
- **Strength of evidence in the most relevant regimes:** The AIME24 addendum shows only modest gains (and only at higher budgets), which undermines the broader claim of “significant” improvements for demanding reasoning where test-time scaling matters most. The explanation (uniformly hard distribution + small model) is plausible but not demonstrated via stronger models during the rebuttal.
- **Sequential rollout vs. latency/parallelism:** Authors acknowledge an efficiency–latency trade-off and suggest batching, but do not provide empirical end-to-end wall-clock results for sequential vs. batched variants (or vs. baselines) under realistic serving constraints. The concern is acknowledged more than resolved.

**Reviewer Scores:**

- **Reviewer t29B (rating 4, marginally below threshold):** Likely **stays at 4**. The added runtime/overhead discussion and promise to move details to main text address their main practical concerns, but does not substantially change their “oversimplified / needs stronger practical demonstration” stance.

- **Reviewer uSZC (rating 4, marginally below threshold):** Likely stays at 4. The AIME24 addition is appreciated, but the gains are modest and the novelty critique is only partially addressed via rewording/clarification.

- **Reviewer cYP5 (rating 2, reject):** Likely moves to 3 (weak reject). The new Easy2Hard-with-trained-predictor comparison directly addresses their hypothesis, and the latency/parallelism concern is acknowledged with a batching discussion, but there is still no strong empirical latency story, so a full jump to “borderline accept” seems unlikely.

- **Reviewer MUor (rating 4, marginally below threshold):** Likely stays at 4. The rebuttal promises clearer positioning, adds missing citations, and gives practical proxy-selection heuristics, but the core critique (incremental contribution, dependence on proxy quality) remains.

---

### Decision · Program_Chairs · 2026-01-26

Reject